# Incidence of Chronic Pulmonary Aspergillosis in Patients with Suspected or Confirmed NTM and TB—A Population-Based Retrospective Cohort Study

**DOI:** 10.3390/jof8030301

**Published:** 2022-03-16

**Authors:** Frederik P. Klinting, Christian B. Laursen, Ingrid L. Titlestad

**Affiliations:** 1Odense Respiratory Research Unit (ODIN), Department of Clinical Research, Faculty of Health Science, University of Southern Denmark, 5000 Odense, Denmark; christian.b.laursen@rsyd.dk (C.B.L.); ingrid.titlestad@rsyd.dk (I.L.T.); 2Department of Respiratory Medicine, Odense University Hospital, 5000 Odense, Denmark

**Keywords:** pulmonary disease, mycobacterium tuberculosis, tuberculosis, non-tuberculous mycobacterium, chronic pulmonary aspergillosis, aspergillosis

## Abstract

Chronic pulmonary aspergillosis (CPA) is a severe and underdiagnosed pulmonary fungal infection with a significant overlap in symptoms and imaging findings of mycobacterium tuberculosis (TB) and non-tuberculous mycobacterium (NTM). Infection with TB or NTM is a predisposing underlying condition for CPA in approximately one-third of patients. A previously published study from Uganda showed increased incidence and complication rate of CPA with respect to pre-existing radiographic cavitation in a post-treatment TB population. The aim of this study was to investigate the incidence of CPA in a low-endemic population of confirmed or suspected TB and NTM patients. We manually reviewed 172 patients referred on suspicion or for treatment of TB or NTM at the Department of Respiratory Medicine, Odense University Hospital during the period of 1 January 2018 to 31 December 2020. We found no CPA amongst TB patients as opposed to an incidence of 8.2% (n = 4) in NTM-infected patients. We identified possible investigatory differences in *Aspergillus* blood sample screening protocols depending on NTM or TB, initiated at the Department of Respiratory Medicine at Odense University Hospital. A focused screening and investigatory protocol in NTM patients with persisting or developing symptoms is warranted in relation to suspected CPA.

## 1. Introduction

Chronic pulmonary aspergillosis (CPA) is a severe and underdiagnosed pulmonary fungal infection with an estimated patient population of up to 3 million people worldwide and a 5-year mortality upwards of 80% or higher [1,2,3,4,5]. In 2015, CPA infections in Denmark were estimated at 4.8 pr. 100,000 persons [6]. The significant overlap in symptoms and imaging findings between mycobacterium tuberculosis (TB), non-tuberculous mycobacterium (NTM), and other respiratory diseases complicates the diagnosis of CPA [7]. As CPA develops in pre-existing or new expanding bronchopulmonary cavities or nodules, it becomes a potentially complicating disease of TB and NTM.

There are only a few international studies on CPA as a complication of TB and NTM [8], and a search of PubMed and Embase Ovid resulted in a few studies concerning low-endemic TB and NTM populations. In an assessment of 126 CPA patients from the U.K., 30.2% had TB or NTM as a predisposing primary underlying condition. Accordingly, immunocompromised individuals and patients with respiratory diseases, such as bronchiectasis, chronic obstructive pulmonary disease (COPD), and multiple comorbidities, have an increased risk of developing CPA [9,10]. Furthermore, a centralised European CPA registry, CPAnet, indicates that approximately 20% of CPA patients have had an active or previous TB or NTM infection prior to CPA [7]. In a recently published prospective study, the authors showed that post-treatment TB patients in Uganda had a CPA complications rate of 4.9–6.3%. Patients with pre-existing radiographic cavitation had a CPA complication rate of 26%, and 71.8% of the CPA diagnosed patients in the study had developed CPA within 5 years from the completion of their TB treatment [1].

Early onset diagnostics, as well as the treatment of CPA, is crucial for the reduction in mortality and improvement in long-term survival in these patients [3]. The aim of this study was to investigate the incidence of CPA in a low-endemic population of referred patients with confirmed or suspected TB and NTM.

## 2. Materials and Methods

### 2.1. Study Design and Setting

The study was performed at the Department of Respiratory Medicine, Odense University Hospital (OUH), Denmark, as a retrospective, non-intervention cohort. The department serves as the primary clinic for the diagnosis and treatment of pulmonary NTM, with patient referrals from all relevant departments in the Region of Southern Denmark, and has the primary function of the assessment and treatment of pulmonary TB on Funen. In 2018, analysis of *Aspergillus*-specific IgE and IgG and total IgE serology was implemented as part of the diagnostic blood testing regimen, as a screening tool for CPA and allergic bronchopulmonary aspergillosis (ABPA) [1,3].

### 2.2. Participant Identification and Data Management

All patients referred for treatment had one or more initiated clinical investigative contacts during the period of 1 January 2018 to 31 December 2020, and one or more of the following International Classification of Disease 10th edition (ICD-10) codes were included in the study:A15*—Mycobacterium tuberculosis infections;A16*—Pulmonary mycobacterium tuberculosis;A31*—Non-tuberculous mycobacterium infection;Z030—Suspicion of tuberculosis.

Data were manually collected at a baseline from electronic medical records, consisting of radiological imaging and serological, microbiological, and pathological results. All data were anonymized upon retrieval from the electronic records. Follow up (FU) was defined as the period from time of referral until death, loss to follow up, or 31 December 2020.

Baseline characteristics were age, sex, body mass index (BMI), medical history, prescription medication, pre-existing medical comorbidities, and symptoms (i.e., cough, dyspnoea, and haemoptysis), and were collected at the time of referral or from record entries resulting in the referral on suspicion of TB or NTM. Radiological patterns were classified according to radiographic descriptions and all radiographic modalities were included if available. All relevant investigative activities and diagnostic tests leading up to the initial clinical contact were included if available, e.g., Aspergillus galactomannan (AGM) or interferon-gamma release assays (IGRA).

TB and NTM diagnoses are subject to mandatory reporting to Statens Serum Institut (SSI) under the Danish Ministry of Health. As such, known exposure to infected TB patients warrants clinical investigatory processes at a hospital despite the lack of symptoms and otherwise suspicion of infection. A clinical consideration can be made during the first clinical contact in order to conduct watchful waiting and refrain from further diagnostic assessments, e.g., radiographic imagery, based upon patient symptoms and presentation.

The variations in individual diagnostic CPA criteria as listed in Section 2.3 were classified as progressive, stable, or regressive patterns during follow up, in order to assess undiagnosed CPA in the population [3,11,12,13]. Diagnostic classification was performed using the ICD-10 system by the clinician during or after the investigative process. Data were stored in a predesigned Microsoft SharePoint database prior to statistical analysis.

### 2.3. Criteria of CPA and CPA Subtyping

A CPA diagnosis was based on European Respiratory Society (ERS) guidelines, modified to accommodate a population containing TB or NTM patients. Patients were diagnosed with CPA if the following conditions were met [3]:Pulmonary symptoms (i.e., cough, sputum, haemoptysis, dyspnoea) persistent for more than 3 months, with or without progression.One or more cavities or nodules characteristic of *Aspergillus* on radiological imaging, with or without progression.Direct or indirect serology evidence, such as *Aspergillus* serology indicative of a relevant immunological response and/or culture, microbiology, or pathological respiratory samples, characteristic of *Aspergillus* infection. Serology was assessed using *Aspergillus* fumigatus IgG and IgE (ImmunoCap, Phadia, Thermo Fischer Scientific, Sweden) with chosen cut-off values of IgG > 75 mg/L, *Aspergillus* fumigatus IgE > 0.35 × 10^3^ IU/L, and total IgE > 115 × 10^3^ IU/L.Exclusion of alternative diagnosis, other than TB or NTM, based on clinical symptoms, radiological patterns, and microbiological evidence and according to Danish national guidelines [12,13].

Patients fulfilling all 4 criteria were diagnosed with CPA and classified as either: Aspergillus nodule, simple aspergilloma, chronic cavitary pulmonary aspergillosis (CCPA), chronic fibrosing pulmonary aspergillosis (CFPA), and subacute invasive aspergillosis (SAIA).

### 2.4. Audit

Medical records were reviewed in post-data-collection audits if one or more criteria of the predefined diagnostic criteria were present or if treatment response, despite relevant treatment, was abnormal and/or a regressive pattern was not established [3]. CPA diagnosis and subtyping was based on bi-consensual agreement between the main author and a medical doctor with relevant TB or NTM and CPA clinical experience.

Patients diagnosed through relevant CPA multi-disciplinary-team (MDT) decisions were not subject to secondary audits. Audited patients with a strong clinical suspicion of *Aspergillus* infection and not fulfilling strict ERS CPA criteria were classified as Possible CPA. Examples of the use of this classification would be absence of *Aspergillus* diagnostic testing, short or incomplete observation time, patients being clinically unresponsive to relevant treatment of underlying illness, absence of anti-aspergillus treatment, or indirect evidence of *Aspergillus* infection.

### 2.5. Primary and Secondary Outcome Variables

The primary outcome of this study was the incidence of CPA in TB and NTM patients. The secondary outcome was the CPA incidence of the included population. The study population was grouped for comparative analysis based upon microbiological evidence of mycobacterium tuberculosis and non-tuberculous mycobacterium or classified as Secondary Diagnosis (SDG); see Figure 1 for further details.

### 2.6. Statistical Analysis

All statistical analysis was performed using IBM SPSS version 27. Quantitative numerical data are presented as means and medians with relevant confidence intervals or a minimum and maximum range using an ANOVA test and independent median sample test, respectively. Fisher’s exact test was used for comparative analysis of groups for categorical variables. All results were tested with regard to a statistical significance of 5%. Missing data were left out of the assessment and statistical analysis.

## 3. Results

In the period of 1 January 2018 to 31 December 2020, a total of 172 patient medical records were reviewed, 52.9% of which belonged to male patients, and the mean age in the population was 52 years. NTM patients had a mean age of 70 years and TB or SDG ages ranged from 42–46. Patients who had been exposed to known TB or NTM infection were included in our setting and, as such, we included children during the study period. These were classified as SDG if not otherwise infected with TB or NTM or diagnosed with latent TB. No patients tested positive for HIV and the mean overall follow up time was 600 days.

In terms of group heterogeneity, pre-existing conditions were unevenly distributed, with 40–43% of NTM patients having COPD or structural, interstitial, and infectious lung disease, as opposed to 5.4–16.3% of TB and SDG patients. Additionally, 44.9% of NTM patients used inhaled corticosteroids (ICS) compared to 2.7% for TB and 12.8% for SDG. For further details, see Figure 1 and Table 1.

In total, seven (4.1%) patients were diagnosed with CPA. In the NTM group, four CPA patients (8.2%) were identified. No patients were diagnosed with CPA in the TB group and three patients (3.5%) were diagnosed amongst the SDG patients. One case of allergic bronchopulmonary aspergillosis (ABPA) and three cases of clinical suspicion of CPA were identified amongst all patients. No statistical significance was found between the groups in a comparative analysis. See Table 2.

In total, 55 (32.0%) patients presented symptoms lasting more than three months at baseline. Nine NTM patients had persistent progressive symptoms during FU (22.4%).

Conventional chest X-rays were conducted in 150 patients during the study period, and additional radiographic modalities (CT, PET/CT, and HRCT) used in the groups significantly varied, with the most comprehensive use in the NTM group. In total, 122 (70.9%) patients had radiographic infiltrate(s) or cavitation(s) at baseline, with 98% of NTM and 100% of TB patients. One in five NTM patients had bronchiectasis (20.4%) and one patient had a fungal ball at baseline. Nineteen (38.8%) patients in the NTM group showed radiological progression, which was significant compared to four (10.8%) in the TB group and six (7.0%) amongst SDG patients.

Testing for aspergillus infections was not consistent in the population. Only 104 (60.5%) patients had aspergillus serology conducted during FU. Approximately 40% of all patients had respiratory samples tested at least once for aspergillus. A total of 77.6% of NTM patients had at least one sample tested and/or cultured for aspergillus during follow up, compared to 59.5% of TB patients and 12.8% among SDG patients. See Table 3 for further diagnostic details.

## 4. Discussion

To the best of our knowledge, we conducted the first CPA study on a combined TB and NTM population in a low-endemic setting.

We could identify a clear difference in the incidence of CPA between the distributed groups. Our findings indicate that patients with NTM infections were more prone to have simultaneous infection with CPA as opposed to SDG and TB patients. We found no evidence of CPA amongst TB patients in our study, which may have coincided with a relatively younger age mean, as well as the extensive control and outreach programs for treating patients with TB in Denmark. Additionally, the issue of CPA and TB coinfections, in a Danish context, is less relevant compared to high-endemic countries. Arising disease complications, such as developing pulmonary cavities post-TB infection, as seen in the Ugandan study [1], may vary due to healthcare availability and socioeconomic and demographic differences in the populations. Furthermore, a difference in general health status and pronounced use of inhalation medications, as seen in our population, may cause a difference of immune response to a potential NTM or TB infection as opposed to an innate immune response.

We found a significant age difference between all groups, with NTM patients being significantly older than those in the other groups. Increased age combined with a higher rate of prolonged symptoms, comorbidities, bronchiectasis, and cavities in NTM patients are all correlating risk factors for CPA [9,14]. Additionally, the use of ICS was more pronounced in the NTM population, which correlated with the significant difference in pre-existing asthma and COPD amongst the NTM patients, compared to the other patient groups, in terms of a high risk of developing CPA [3,4,14,15].

NTM is often found in the assessment of infiltrates, mimicking malignancy, potentially resulting in a subsequent referral to the Centre for Tuberculosis. As a result, most NTM patients included in our study had more extensive testing and radiology conducted at the time of referral compared to the other groups. As such, a difference in the diagnostic measures conducted at baseline in our study may have led to underdiagnosed CPA in the population, especially with respect to TB and SDG.

The chosen *Aspergillus fumigatus* IgG threshold was based on local laboratory recommendations, but an approach using some of the lower cut-off values previously reported in the literature could also have been used. A consensus regarding Aspergillus fumigatus IgG and IgE serology is still pending. Some studies and guidelines advocate for lower IgE and IgG cut-off values, as low as >10 mg/L, which may result in a higher prevalence of false-positive CPA patients, as opposed to a higher, conservative cut-off value, e.g., 75 mg/L, which may lead to an underestimation of CPA patients. However, ERS guidelines stipulates and emphasizes that; “*Aspergillus IgG serology do not allow a definitive conclusion about comparative diagnostic performance for CPA*”; hence, it should be compared to clinical patient presentation and the need for further investigatory exploration [3] (p. 53).

Some studies suggest that most cases of aspergillus infection in NTM patients reflect colonisation without an increase in mortality, while others stipulate that simultaneous CPA and NTM infections are indicative of a higher mortality risk [4,16,17]. Our observation period of 20 months on average showed no relapses of TB or NTM or post-treatment-diagnosed CPA and did not distinguish between CPA colonisation or clinical infection in need of treatment. Furthermore, we found no significant difference in mortality between the groups during our study period. Three patients were classified with Possible CPA during post-data-collection audits due to inconsistency in aspergillus testing. ERS guidelines stipulate that extensive radiology examinations and invasive diagnostics such as bronchoscopy specimens, biopsies, and PCR analysis of respiratory samples are better diagnostic tools for aspergillus infection [3]. Patients with regressive symptoms after initiating relevant treatment warranted no clinical suspicion of CPA; hence, they had no further diagnostic testing, which could justify the difference in aspergillus tests conducted.

Additional or increased screening during the investigative process in order to identify potentially undiagnosed CPA infections could involve systematic aspergillus serology, radiology, and invasive diagnostics during treatment of TB or NTM. Furthermore, an increased observation time in order to clarify post-treatment complications and outcome should be considered. However, increased screening efforts should result in a clinical treatment option and improved clinical outcomes for patients. In a low-endemic setting and with a relatively low CPA incidence of 4.1%, our results indicate that increased screening to some extent is warranted in patients diagnosed with NTM, having a higher prevalence of CPA (8.2%), even if some CPA cases might not be identified among other patient groups. Pre-existing risk factors such as structural lung tissue damage and the resulting risk of opportunistic lung infection in NTM patients should raise clinical suspicion and lower the investigatory threshold for CPA when presenting with persisting or developing pulmonary symptoms after treatment.

It remains uncertain what effect additional screening would have in patients with NTM, due to pre-existing comorbidities and/or adverse effects to long-term antibiotic treatment, in terms of outcome, mortality, and health economics, which could be investigated further in prospective studies.

### 4.1. Limitations

Using a retrospective model and only including patients from a single centre provided a small study population. The inclusion criteria were dependent on accurate diagnostic coding by clinicians, which may have excluded a proportion of the population. Therefore, comparative analysis of groups may be inadequate and subject to selection bias.

Data collection was conducted with a clinical approach and included all patients suspected of TB or NTM. However, the involvement of regional departments of medicine dispersed across the Region of Southern Denmark, as well as being confined to existing data, may have resulted in incomplete data and potentially undiagnosed CPA. Furthermore, the inconsistent aspergillus testing regimen amongst a proportion of the patients may result in false-negative CPA patients due to a lack of investigatory activities. Hence, the number of CPA-infected patients may be larger than we were able to prove in this population.

Radiological data were classified based on radiographic descriptions in medical records and were only subject to secondary audits if indicative of CPA.

### 4.2. Generalisability

Most studies published on TB, NTM, and CPA have been conducted in a high-endemic setting in either Africa, Asia, or the Middle East, making direct comparison difficult. Previous low-endemic studies have been conducted in Manchester, U.K., where access to national healthcare services and socioeconomic status is not directly comparable to a Danish context [4,9]. However, our results indicate that continuous CPA screening in patients with NTM in a low-endemic setting is potentially relevant for countries, e.g., Europe or other settings comparable to Denmark.

## 5. Conclusions

CPA coinfections should be considered by the clinician in a low-endemic TB setting when NTM patients present with persisting or developing pulmonary symptoms despite relevant treatment. *Aspergillus fumigatus* IgG can be used as an initial screening method in NTM patients; however, comprehensive supplementary radiographic imaging, serology, microbiological testing, and invasive procedures need to be considered in order to ensure an accurate CPA diagnosis and subtyping.

## Figures and Tables

**Figure 1 jof-08-00301-f001:**
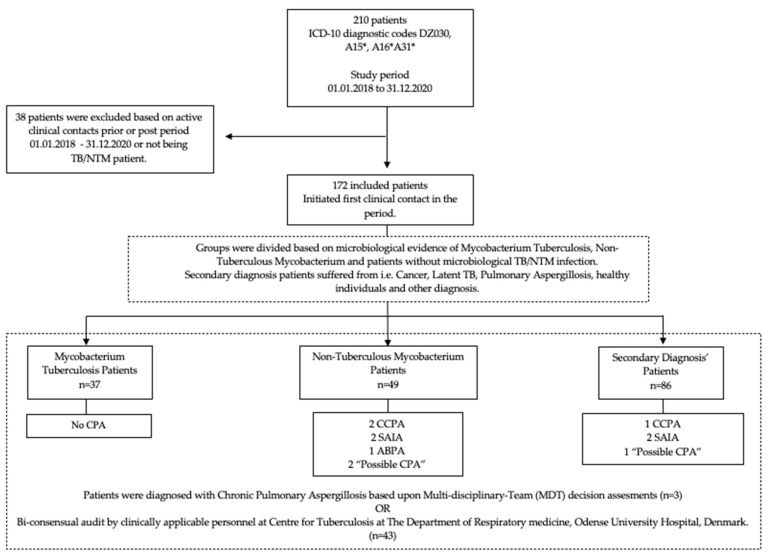
Flowchart of study population.

**Table 1 jof-08-00301-t001:** Population characteristics.

	All Patients	Non-Tuberculous Mycobacterium Patients	Mycobacterium Tuberculosis Patients	Secondary Diagnosis Patients	*p*-Value
Subjects (n)	172 (100)	49 (28.5)	37 (21.5)	86 (50.0)	
Male sex	91 (52.9)	22 (44.9)	25 (67.6)	44 (51.2)	0.096
Age (years) mean (min; max)	52 (0; 88)	70 (26; 86)	42 (18; 62)	46 (0; 88)	<0.001
BMI (kg/m^2^) median	22.19 (21.68–22.60)	21.84 (20.28–22.49)	22.28 (21.22–23.55)	22.6 (21.68–24.50)	0.566
Alcohol > 14 (male sex) > 7 (female sex) pr. week	17 (9.9)	3 (6.1)	9 (24.3)	5 (5.8)	0.008
Smoking (active)	58 (33.7)	14 (28.6)	22 (59.5)	22 (25.6)	<0.001
Substance abuse (current)	9 (5.2)	1 (2.0)	3 (8.1)	5 (5.8)	0.393
Substance abuse (prior)	5 (2.9)	3 (6.1)	1 (2.7)	1 (1.2)	0.180
**Comorbidities**					
Asthma	14 (8.1)	8 (16.3)	1 (2.7)	5 (5.8)	0.054
COPD	27 (15.7)	20 (40.8)	2 (5.4)	5 (5.8)	<0.001
Structural, interstitial and infectious lung disease	37 (21.5)	21 (42.9)	2 (5.4)	14 (16.3)	<0.001
Diabetes mellitus	13 (7.6)	7 (14.3)	2 (5.4)	4 (4.7)	0.159
Immunosuppressive disease	3 (1.7)	3 (6.1)			0.031
Autoimmune/rheumatic disease	14 (8.1)	8 (16.3)	1 (2.7)	5 (5.8)	0.054
Cancer former/current	25 (14.5)	12 (24.5)	1 (2.7)	12 (14.0)	0.013
No known illnesses	35 (20.3)	1 (2.0)	11 (29.7)	23 (26.7)	<0.001
**Medications**					
Inhaled corticosteroid	34 (19.8)	22 (44.9)	1 (2.7)	11 (12.8)	<0.001
Systemic steroid	10 (5.8)	4 (8.2)		6 (7.0)	0.216
Immunosuppressants (non-steroid)	8 (4.7)	1 (2.0)		7 (8.1)	0.112
Diabetics medication	11 (6.4)	7 (14.3)	1 (2.7)	3 (3.5)	0.041
Antibiotics use chronic/intermittent	7 (4.1)	4 (8.2)	2 (5.4)	1 (1.2)	0.085
No medicine at baseline	32 (18.6)	2 (4.1)	9 (24.3)	21 (24.4)	0.003

Data are presented as n (%) unless otherwise stated. ANOVA test applied to means and medians, tested using independent samples median test. All patients tested for HIV were negative. COPD: chronic obstructive pulmonary disease. Structural, interstitial, and infectious lung disease: previous TB or NTM infection, chronic pulmonary infections, e.g., Pseudomonas aeruginosa colonisation, thoracic surgery, bronchiectasis, lung fibrosis, and other. Inhaled corticosteroid: including single and/or combination medications. Diabetic medications, e.g., insulin or GLP1-analog, etc. Diabetes Mellitus (DM): unspecified DM, DM 1, DM2 and associated complications including DM-induced retinopathy, glomerulopathies, etc.

**Table 2 jof-08-00301-t002:** Incidence of chronic pulmonary aspergillosis.

	All Patients	Non-Tuberculous Mycobacterium Patients	Mycobacterium Tuberculosis Patients	Secondary Diagnosis Patients	*p*-Value
Subjects (n)	172 (100)	49 (28.5)	37 (21.5)	86 (50.0)	
CCPA	3 (1.7 (0.5–4.6))	2 (4.1 (0.9–12.5))	-	1 (1.2 (0.1–5.3))	0.313
SAIA	4 (2.3 (0.8–5.4))	2 (4.1 (0.9–12.5))	-	2 (2.3 (0.5–7.3))	0.669
CPA	7 (4.1 (1.8–7.8))	4 (8.2 (2.8–18.2))	0 (0 (0.92–1)) ^a^	3 (3.5 (1.0–9.0))	0.171
Possible CPA	3 (1.7 (0.5–4.6))	2 (4.1 (0.9–12.5))	-	1 (1.2 (0.1–5.3))	0.313
ABPA	1 (0.6 (0.1–2.7))	1 (2.0 (0.2–9.1))	-	-	0.500

Data are presented as n (%) unless otherwise stated. CPA; chronic pulmonary aspergillosis, CCPA; chronic cavitary pulmonary aspergillosis, SAIA; subacute invasive aspergillosis, ABPA; allergic bronchopulmonary aspergillosis. CPA: includes subtype CPA, i.e., CCPA and SAIA. Clinical suspicion of CPA: author definition of strong suspicion of Aspergillus infection. ^a^ Calculated using the rule of three, with respect to *p* < 0.05.

**Table 3 jof-08-00301-t003:** Diagnostic and investigatory parameters.

	All Patients	Non-Tuberculous Mycobacterium Patients	Mycobacterium Tuberculosis Patients	Secondary Diagnosis Patients	*p*-Value
Subjects (n)	172 (100)	49 (28.5)	37 (21.5)	86 (50.0)	
Observational FU time (days) Mean n (95CI)	600 (556–645)	541 (458–624)	508 (418–599)	673 (610–737)	0.004
FU time–clinical duration (days) Mean n (95CI)	316 (278–354)	405 (327–483)	345 (300–391)	253 (197–309)	0.002
**Symptoms at baseline**					
Symptom duration > 3 months	55 (32.0)	16 (43.2)	23 (46.9)	16 (18.6)	<0.001
Dyspnoea	47 (27.3)	6 (16.2)	28 (57.1)	13 (15.1)	<0.001
Cough	110 (64.0)	27 (73.0)	35 (71.4)	48 (55.8)	0.091
Fever	21 (12.2)	7 (18.9)	6 (12.2)	8 (9.3)	0.346
Lymphadenitis	4 (2.3)	2 (5.4)		2 (2.3)	0.254
Haemoptysis	19 (11.0)	7 (18.9)	7 (14.3)	5 (5.8)	0.057
Night sweats	14 (8.1)	5 (13.5)	3 (6.1)	6 (7.0)	0.415
Weight loss (unwanted/pathological > 5%/6 months)	38 (22.1)	20 (54.1)	11 (22.4)	7 (8.1)	<0.001
Non-pulmonary/other symptoms	78 (45.3)	25 (67.6)	24 (49.0)	29 (33.7)	0.002
Asymptomatic	28 (16.3)	1 (2.7)	3 (6.1)	24 (27.9)	<0.001
**Symptoms during FU**					
Progressive symptoms during FU	12 (7.0)	11 (22.4)		1 (1.2)	<0.001
Unchanged symptoms during FU	42 (24.4)	7 (14.3)	27 (73.0)	32 (37.2)	<0.001
**Aspergillus serology**					
Aspergillus serology at baseline	90 (52.3)	29 (59.2)	29 (78.4)	32 (37.2)	<0.001
Aspergillus serology during FU	14 (8.1)	5 (10.2)	2 (5.4)	7 (8.1)	0.766
IgE > 115 × 10^3^ IU/L	42 (24.7)	14 (28.6)	16 (43.2)	12 (14.3)	0.002
Asp. IgE > 0.35 × 10^3^ IU/L	6 (3.5)	5 (10.2)		1 (1.2)	0.020
Asp. IgG > 75 mg/L	14 (8.1)	8 (16.3)	3 (8.1)	3 (3.5)	0.054
**Aspergillus histology/cytology**					
Respiratory sample tested for Aspergillus (any time)	71 (41.3)	38 (77.6)	22 (59.5)	11 (12.8)	<0.001
Positive aspergillosis test at baseline	7 (4.1)	4 (8.2)	1 (2.7)	2 (2.3)	0.267
Positive aspergillosis test during FU	7 (4.1)	4 (8.2)		3 (3.5)	0.171
BAL/BL/Serum AGM positive	8 (4.7)	3 (6.1)	1 (2.7)	4 (4.7)	0.803
**Baseline Radiology**					
Radiology > 1 cavity/nodule/infiltrate at baseline	122 (70.9)	48 (98.0)	37 (100)	37 (43.0)	<0.001
Bronchiectasis	20 (11.6)	10 (20.4)	4 (10.8)	6 (7.0)	0.062
Aspergilloma/fungal ball	1 (0.6)	1 (2.0)			0.500
Cavity/abscess without fungus	28 (16.3)	10 (20.4)	14 (37.8)	4 (4.7)	<0.001
Unspecified consolidated/cavitating radiological abnormality	87 (50.6)	38 (77.6)	25 (67.6)	24 (27.9)	<0.001
Nodule < 30 mm	50 (29.1)	20 (40.8)	18 (48.6)	12 (14.0)	<0.001
Mass > 30 mm	15 (8.7)	7 (14.3)	7 (18.9)	1 (1.2)	<0.001
Diffuse infiltrative changes	21 (12.2)	10 (20.4)	7 (18.9)	4 (4.7)	0.006
No baseline infiltrates	48 (27.9)	1 (2.0)		47 (54.7)	<0.001
Pleural abnormalities	42 (24.4)	18 (36.7)	9 (24.3)	15 (17.4)	0.049
**Radiological modalities**					
Chest X-ray	150 (87.2)	41 (83.7)	35 (94.6)	74 (86.0)	0.282
CT	58 (33.7)	24 (49.0)	17 (45.9)	17 (19.8)	<0.001
PET/CT	39 (22.7)	15 (30.6)	18 (48.6)	6 (7.0)	<0.001
HRCT	18 (10.5)	15 (30.6)		3 (3.5)	<0.001
No baseline radiology	5 (2.9)			5 (5.8)	0.131
No FU radiology	36 (20.9)	5 (10.2)	1 (2.7)	30 (34.9)	<0.001
Radiology progression during FU	29 (16.9)	19 (38.8)	4 (10.8)	6 (7.0)	<0.001
Anti-tuberculosis treatment during FU	97 (56.4)	35 (71.4)	37 (100)	25 (29.1)	<0.001
Anti-fungal treatment during FU	6 (3.5)	3 (6.1)		3 (3.5)	0.408
Average treatment duration (days) Median	149	266 (188–343)	218	52 (34–70)	<0.001
Deaths during FU	11 (6.4)	6 (12.2)		5 (5.8)	0.503
**NTM microbiology–species ***					
*M. abscessus*	1	1			
*M. avium*	22	20		2	
*M. celatum*	1	1			
*M. chelonae*	1	1			
*M. chimaera*	12	12			
*M. gordonae*	2	2			
*M. heckeshornense*	1	1			
*M. intracellulare*	2	2			
*M. kansasii*	2	2			
*M. malmoense*	1	1			
*M. neoarum*	1	1			
*M. parascrofulaceum*	1	1			
*M. szulgai*	1	1			
*M. xenopi*	2	2			

Data are presented as n (%) unless otherwise stated. ANOVA test applied to means and medians, tested using independent samples median test. * NTM microbiology–species includes all registered microbiological evidence recorded during the study period. AGM: Aspergillus galactomannan. Aspergillus serology: serology test incl. Aspergillus Fumigatus IgE, IgG and unspecified IgE. CT: computer tomography, PET/CT: positron emission tomography, HRCT: high-resolution CT, FU: Follow up. Pleural abnormalities: pleural thickening, effusion, calcification, and empyema at baseline.

## Data Availability

Data can be made available upon request.

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
