# Peer review of "Incidence of Chronic Pulmonary Aspergillosis in Patients with Suspected or Confirmed NTM and TB—A Population-Based Retrospective Cohort Study"

_jof, 2022, doi:10.3390/jof8030301_

Round 1

Reviewer 1 Report

Even taking into account the study limitations, as recognized by the authors in item 4.1 (retrospective design, single-center, inclusion criteria dependent on accuracy diagnostic coding by clinicians, and inconsistent aspergillus testing), I consider the present findings crucial for understanding the conundrum of the association of CPA with TB/NTM. 

Author Response

No comments were provided by this reviewer, hence point-by-point responses have not been conducted for reviewer 1.

Reviewer 2 Report

The article is concise and well-written. The authors explain the methods in a systematic and great details about the detection of CPA in NTM and TB populations. However, there are some issues that require additional confirmations. Specific comments follow.

  1. Page 2: Study design and setting

The department serves as the primary clinic for treatment of pulmonary NTM with referrals of patients from other departments in Region of Southern Denmark and has the primary function for assessment and treatment of pulmonary TB on Funen.

Do you think that the fact the hospital is the referral of NTM might bring biases to the NTM population? The patients that come to OUH will be the complicated cases of NTM rather than TB patients.

As we see in table 1, the comorbidities are always higher in NTM population compare to TB.

  1. Page 2: The variation of individual CPA criteria during follow-up, were classified as progressive, stable or regressive patterns in order to asses undiagnosed CPA in the population [3,11–13].

Assess instead of asses.

Please explain in the result what is the percentage of progressive, stable or regressive CPA in Results/Discussion.

  1. Could you provide the p-value in the tables?
  2. Page 4: These were classified SDG if not otherwise infected with TB/NTM or diagnosed with latent TB.

How many patients had latent TB?

  1. Page 6: Conventional chest X-rays was conducted in 150 patients during the study period, and additional radiographic modalities (CT, PET/CT, and HRCT) used in the groups varied significantly, with most comprehensive use in the NTM group.

The number of patients is 172, but only 150 have radiology tests?

How did you diagnose CPA in the patients with no radiology information?

  1. Could you please add the number of patients in every population for table 3?

Did all 172 patients at baseline were included in the follow up?

  1. Page 9: Furthermore, we found no significant difference in mortality between the groups during our study period.

Please add the mortality rate in each group, probably in the table or texts.

  1. Could you add the information about the species of NTM detected in your patients?

Author Response

Dear editor

Thank you for the comments to our manuscript entitled : Incidence of chronic pulmonary aspergillosis in patients with suspected or confirmed non-mycobacterial and mycobacterial tuberculosis – A population-based retrospective cohort study”. You have very kindly given us advice on how to clarify and revise the paper. All reviewer comments are addressed below in red.

POINT 1

Page 2: Study design and setting

The department serves as the primary clinic for treatment of pulmonary NTM with referrals of patients from other departments in Region of Southern Denmark and has the primary function for assessment and treatment of pulmonary TB on Funen.

Do you think that the fact the hospital is the referral of NTM might bring biases to the NTM population? The patients that come to OUH will be the complicated cases of NTM rather than TB patients.

Answer:

As the single center at OUH handles the entire population, both complicated and uncomplicated, of NTM in a fairly small danish population, this provides the best possibility to examine the NTM population of the region.

Furthermore, all NTM patients, regardless of severity, are referred for treatment at OUH, resulting in a complete inclusion of the regions NTM patients in our study during the period.

As danish patients have equal rights and access to health services across the nation/region, patients presenting with relevant symptoms will be examined and have relevant testing conducted. However, as such we cannot reject the fact that, potentially underdiagnosed asymptomatic NTM and CPA co-infected patients exist in the region without contact to healthcare services. However, these require no urgent treatment.

The TB population of Funen may underestimate the national CPA and TB co-infected population, however severe and complicated TB patients are referred from the Region of Southern Denmark if needed in order to treat aspergillus, CPA or other coinfections at OUH, as this is the regional university hospital for specialist treatment.

Relevant for both populations is, that the population in total in the region of southern Denmark is limited as TB/NTM is considered a potentially severe disease in Denmark subject to special surveillance; however the total infected population is small nationally.

The section has be adjusted slightly to make this more clear. See page 2 line 59-60.

POINT 2

As we see in table 1, the comorbidities are always higher in NTM population compare to TB.

Page 2: The variation of individual CPA criteria during follow-up, were classified as progressive, stable or regressive patterns in order to asses undiagnosed CPA in the population [3,11–13].

Assess instead of asses.

Answer:

A spelling correction has been made.

POINT 3

Please explain in the result what is the percentage of progressive, stable or regressive CPA in Results/Discussion.

Answer:

The individual criteria in order to diagnose a patient with CPA were classified as progressive, stable and regressive patterns as a result of e.g. developing of pulmonal symptoms despite relevant TB/NTM treatment during follow-up. We did not classify the CPA diagnosis by progression, stability or regression during follow-up as commented on by reviewer.

In table 3, we have disclosed results concerning progressive and unchanged radiological patterns as well as symptoms. Furthermore, the baseline and follow-up prevalence of aspergillus serology, histology and cytology has been disclosed as wel in the results sectionl. Please see page 6 line 203-218.

Regressions in symptoms, radiology, lack of microbiology/cytology and histology is less relevant to present in results and tables, as these imply that relevant treatment were initiated during the investigatory period and warranted no suspicion of CPA in these patients.

The section 2.2 has be adjusted slightly to make this more clear. See page 2 line 94-96.

POINT 4

Could you provide the p-value in the tables?

Answer:

The p-value could be provided and has previously in the process been discussed amongst the authors to be provided in the tables. We decided that, based on population size, that the use of“ * (p-value <0.05) “ as well as the 95% CI for statistically significant results would describe the comparative analysis between the three groups in a more sufficient way. Per request of the reviewer, the p-value has been added to all tables.

POINT 5

Page 4: These were classified SDG if not otherwise infected with TB/NTM or diagnosed with latent TB.

How many patients had latent TB?

Answer:

Overall data collection was based upon ICD10-codes. Patients were afterwards grouped based on ICD10 coding as well as medical records review throughout the study period. However, some patients were miscoded  and were as such allocated to SDG based according to treatment regime and clinical considerations upon medical record review during data-collection.

If patients had no relevant pulmonary symptoms, no microbiological evidence of TB/NTM, were diagnosed with other illnesses and had no need for further treatment during the follow up, they were grouped as SDG during data collection.  As such we do not have a specific grouping for latent TB patients, and this cannot be provided. See figure 1.

POINT 6

Page 6: Conventional chest X-rays was conducted in 150 patients during the study period, and additional radiographic modalities (CT, PET/CT, and HRCT) used in the groups varied significantly, with most comprehensive use in the NTM group.

The number of patients is 172, but only 150 have radiology tests?

How did you diagnose CPA in the patients with no radiology information?

Answer:

The population of 172 patients included patients suspected of and/or confirmed infected with TB/NTM based upon inclusion and exclusion criteria. The study has included persons with TB exposure, and if the TB-skintest or IGRA was negative, and the persons did not have symptoms, there was no further assessment according to national standards.

As TB and NTM is considered severe diseases of special status in Denmark, all patients diagnosed are subject to mandatory reporting to Statens Serum Institut (SSI) under the Danish Ministry of Health. As such known exposure to infected persons warrant clinical investigatory processes at the hospital despite the lack of symptoms and otherwise suspicion of infection. A clinical consideration can be made, during the first clinical contact, in order to conduct watch-full waiting and refrain from having radiographic imagery conducted according to national health standards. This explain why only 150 patients had conventional X-rays conducted during the study period, as we included the entire population of first initial clinical contacts during the study period which includes latent TB as well as non-infected individuals.

An elaborating  paragraph has been added to section 2.2 page 2, line 86-93.

POINT 7

Could you please add the number of patients in every population for table 3?

Answer:

Patients pr. group in the population is placed in every table on the first row.

POINT 8

Did all 172 patients at baseline were included in the follow up?

Answer:

As demonstrated in figure 1 on page 4, 172 patients were included according to the inclusion criteria – ICD10, study period, initial clinical contact etc. 38 patients were excluded prior to data collection due to lack/miscoding of relevant ICD10 and first initial contact prior to study period.

The 172 patients were reviewed according to the principles described in section 2.2. page 2, line 66. Follow-up (FU) was defined as the period from time of referral until death, loss-to-follow-up, or December 31st, 2020 for all included in the study.

POINT 9

Page 9: Furthermore, we found no significant difference in mortality between the groups during our study period.

Please add the mortality rate in each group, probably in the table or texts.

Answer:

The recorded deaths are disclosed in table 3 page 8. The Cause of death is however not included, as we did not have permission to access the Death Registry of Denmark.

POINT 10

Could you add the information about the species of NTM detected in your patients?

Answer:

The underlying NTM species have been added to table 3 as requested. See page 8 - 9.

We hope that the corrections are sufficient, and we look forward to be hearing from you.

Yours sincerely

on the behalf of the authors

Frederik Pors Klinting

Department of Respiratory Medicine, Odense University Hospital, Denmark.

Institute of Clinical Research, University of Southern Denmark, Denmark.